# Productive Performance, Serum Antioxidant Status, Tissue Selenium Deposition, and Gut Health Analysis of Broiler Chickens Supplemented with Selenium and Probiotics—A Pilot Study

**DOI:** 10.3390/ani12091086

**Published:** 2022-04-22

**Authors:** Shengting Deng, Shengjun Hu, Junjing Xue, Kaili Yang, Ruiwen Zhuo, Yuanyuan Xiao, Rejun Fang

**Affiliations:** 1College of Animal Science and Technology, Hunan Agricultural University, Changsha 410128, China; dengshengting97@163.com (S.D.); hsjtais@163.com (S.H.); xuejunjing163@163.com (J.X.); kellyyang94@126.com (K.Y.); zhuoruiwen97@163.com (R.Z.); 18374841607@163.com (Y.X.); 2Hunan Co-Innovation Center of Animal Production Safety, Changsha 410128, China

**Keywords:** selenium, probiotics, antioxidant capacity, gut health, broiler chicken

## Abstract

**Simple Summary:**

For several years, the benefits of dietary selenium or probiotics on growth and development of broilers have been reported. However, the effect of a combination of different forms of selenium and probiotics on broilers remains to be seen. Thus, the hypothesis of this study was that supplementing selenium and probiotics have positive synergistic effects and interactions on growth performance, antioxidant capacity, tissue selenium deposition, and intestinal health of broilers. The results of this study demonstrate that selenium and probiotics have positive interactions on the tissue selenium content, duodenum, and jejunum development, as well as the composition of cecum microbial, and there were no significant interactions on the growth performance, and antioxidant capacity. Overall, our findings suggest that the combination of organic selenium and probiotics is superior to inorganic or organic selenium alone in poultry production.

**Abstract:**

The effect and interaction of dietary selenium (Se) and probiotics on three yellow chicken growth performance, tissue Se content, antioxidant capacity, and gut health were studied from 0 to 70 days of age. A total of 400 one-day-old broilers were distributed into four groups (I-Se, O-Se, I-Se + pros, and O-Se + pros groups) consisting of a 2 × 2 factorial design. The main factors were the source of Se (I-Se = inorganic Se: 0.2 mg/kg sodium selenite; O-Se = organic Se: 0.2 mg/kg Selenium yeast) and the level of probiotics (0.5% EM or 0% EM, the component of EM mainly includes *Lactobacillus* and *Yeast* at the dose of 2 × 10^8^ cfu/kg and 3 × 10^7^ cfu/kg, respectively). Each treatment had 5 duplicates consisting of 20 broilers. The results showed that the I-Se group had a greater (*p* < 0.05) ratio of feed: weight gain (F/G) of broilers at Starter (0–35 d) than the other treatments. Compared to the I-Se group, the O-Se group increased (*p* < 0.05) Se concentrations in the liver, pancreas, breast muscles, thigh muscle, and the activity of total antioxidative capacity (T-AOC) in serum, as well as the relative abundance of *Barnesiella* and *Lactobacillus* in cecum. Meanwhile, probiotics enhanced (*p* < 0.05) Se concentrations in the pancreas, thigh muscle, serum, and the activity of T-AOC and glutathione peroxidase (GSH-Px), the duodenum’s ratio of villi height to crypt depth (V/C), the jejunum villus height and V/C, and the ileum’s villus height. Furthermore, the significant interactions (*p* < 0.05) between Se sources and the level of probiotics were observed in Se concentrations in the pancreas, thigh muscle, serum, crypt depth of duodenum, and villus height of jejunum of birds, and *Barnesiella* abundance in the cecal. In conclusion, our results demonstrate that the combination of O-Se + pros can improve broiler early growth performance, tissue Se content in the pancreas, thigh muscle, and serum, promote intestinal development, and regulate the composition of intestinal flora, suggesting a better combination. These findings provide an effective method of nutrient combination addition to improving the performance of three yellow chickens.

## 1. Introduction

Three yellow chicken is one of the indigenous famous breeds in China, which has a huge breeding population, but the related research is insufficient. Selenium is regarded as a trace element of significant physiological relevance for developing chicks because it engages in several processes, including antioxidant response and hormone release [1]. Se insufficiency is associated with decreased poultry growth and reproductive function [2]. In recent decades, there has been a lot of research on broiler sensitivity to dietary Se levels and Se sources [3,4]. As a component of several selenoproteins, Se can serve in the maintenance of antioxidant defenses and the prevention of tissue damage [5]. Generally, Se is supplemented in the diet with inorganic salts (such as sodium selenite) or organic Se-enriched yeast [6]. However, as compared with inorganic forms, organic Se has a higher absorption and retention rate [7]. Because yeast Se is an organic structure, it is less toxic than sodium selenite, easier to digest, and has higher retention and bioavailability [8,9]. Furthermore, due to the distinct absorption routes of organic Se, organic Se has a greater retention rate in muscle tissue than inorganic Se. Compared to the inorganic Se, faster levels of organic Se retained in the spleen, duodenum, and ileum imply a higher rate of Se absorption [10]. Organic Se produced from yeast had greater Se content in liver and breast tissues than elemental Se, sodium selenite, and basal diets in birds [11]. Moreover, recent research has shown that Se-enriched yeast can prevent intestinal damage in broilers while also dramatically increasing body weight, feed coefficient, villus height, and villus/crypt ratio [12]. 

Probiotics are non-pathogenic bacteria that may resist digestion in the host’s gastrointestinal tract and survive in the colon, where they benefit the host’s health [13]. Many studies have found that administering probiotics to farmed commercial animals can improve feed conversion rate (FCR), weight gain, egg–milk production, and reduce morbidity and mortality [14,15,16]. Meanwhile, experts have been exploring effective microorganisms (EM) in broiler production for several years. EM is composed of different microbes, including photosynthetic bacteria, actinomycetes, yeast, lactobacillus, and fungi [17]. They can reduce the incidence of diarrhea, lower the pH of the gut, and therefore minimize the ideal circumstances for the development of harmful bacteria when given to feed or water [18]. EM probiotic spray or spray application is used to clean poultry shelters on a hygienic and biological level [19]. EM probiotics have recently been used to improve poultry production [18,20,21,22]. 

Previous studies have shown that selenium and probiotics have similar positive effects on improving performance and intestinal health of broilers [8]. However, no studies have been conducted on the effects of their combination on broiler chickens. Thus, the current study aimed to investigate the effects and interaction of organic Se or inorganic Se plus probiotics on broiler chicken growth performance, antioxidant capacity, tissue Se deposition, and intestinal health.

## 2. Materials and Methods

### 2.1. Animals, Diets, and Experimental Design

A total of 400 one-day-old healthy three yellow chickens (male, average initial weight was 35.72 ± 0.05 g) were randomly distributed into four groups (I-Se, O-Se, I-Se + pros, and O-Se + pros groups) consisting of a 2 × 2 factorial design. The main factors were the source of Se (I-Se = inorganic Se: 0.2 mg/kg sodium selenite; O-Se = organic Se: 0.2 mg/kg Selenium yeast) and the level of probiotics (0.5% EM or 0% EM). Each groups had 5 duplicates consisted of 20 broilers. The basal diet was designed to meet the nutritional needs of broilers as stated by the National Research Council (NRC) (1994), and the feeding schedule included Starter (0–35 day) and Finisher (36–70 day) diets (Table 1). The additional level of microelement was according to our previous study [23]. The broiler chickens in the I-Se and I-Se + pros groups were fed the basal diet supplemented with 0.2 mg/kg sodium selenite, and in the O-Se and O-Se + pros groups were fed the basal diet supplemented with 0.2 mg/kg Selenium yeast (selenium content is 0.2%, Xingjia Biological Engineering Co., LTD, Changsha, China). Meanwhile, the I-Se + pros and O-Se + pros groups were supplemented with 0.5% EM (Bokaxi, Changsha, China) which mainly includes *Lactobacillus* and *Yeast* at the dose of 2 × 10^8^ cfu/kg and 3 × 10^7^ cfu/kg, respectively.

### 2.2. Growth Performance Measurement

The trial lasted for 70 days. The birds had ad libitum access to feed and water. Birds were weighed in the morning at 1, 35, and 70 d, and feed intake was recorded each day. At the end of the trial, body weight (BW), average daily feed intake (ADFI), average daily gain (ADG), and feed conversion ratio (F/G) were calculated.

### 2.3. Sample Collection

At the end of the experiment, 2 chickens from each replicate (10 per treatment) were selected and fasted for 24 h before slaughter, and electrically stunned and euthanized by cervical dislocation. Blood samples of 20 mL were collected from the jugular vein, and blood vessels were collected obliquely. After centrifuging the supernatant at 3000× *g* r/min for 10 min, 2–4 mL of supernatant was absorbed with a pipet gun and injected into 0.5 mL centrifuge tubes. The liver, kidney, pancreas, breast muscles, thigh muscles, and cecum contents samples were immediately collected and stored at −80 °C in 2 mL cryopreservation tubes. Sections of the intestine (duodenum, jejunum, and ileum) were collected and fixed in 4% paraformaldehyde before being treated with a graded series of ethanol (70, 96, and 100%) to remove water. The tissues were mixed with paraffin before being mounted on slides in 5-mm sections. The slides were stained with hematoxylin and eosin and analyzed under a microscope at a magnification of 100×.

### 2.4. Detection of Tissue Se Content 

The selenium content in the solution was determined following the procedure of Hydride atomic fluorescence spectrometry (GB/T 13883-2008, China), Briefly, 2.0 g of liver, kidney, pancreas, chest muscle, thigh muscle, and 1 mL of the serum sample were digested in a triangular flask containing 25 mL of 4:1 mixture of nitric and perchloric acid for 24 h. The solution was mixed and subsequently digested in a microwave digestion system. The mixtures were heated on the adjustable electric heating plate at 180 °C to leave 2 mL of solution, and deionized water was added to the solution to produce a volume of 10 mL. The colorless mixture was supplemented with 5 mL of ultra-pure water and heated again to discharge nitric acid. When the digestion mixture cooled down, it was placed in a 25-mL volumetric flask to determine the volume. The calibration of the final digestion mixture must be within the calibration range. The mixture was placed at 4 °C without light for detection. The Se content was determined using an atomic fluorescence photometer after the samples were retreated (AFS-920, Titan, Beijing, China). All specimens were analyzed three times with known standards to ensure reproducibility of the results.

### 2.5. Serum Antioxidant Capacity Analysis

The total antioxidant capacity (T-AOC), malondialdehyde (MDA), superoxide dismutase (SOD), and glutathione peroxidase (GSH-Px) were determined strictly according to the operating instructions of the kit purchased from Nanjing Jiancheng Institute of Biological Engineering (Nanjing, China).

### 2.6. Intestinal Morphometry and 16S RNA-Based Microbiota Analysis

At 70 days of age, the gut morphology of the duodenum, jejunum, and ileum was evaluated by measuring the villus length (VL), crypt depth (CD), and villus length/crypt ratio (VL/CD). The samples were prepared following Bai et al. with minor adjustments [24]. The measurements were made in whole slides stained with hematoxylin and eosin (H&E) using a Leica DM500 photomicroscope (Leica Microsystems, Heerbrugg, Switzerland) with 4X objective and the help of the Leica LAS EZ software (Leica Application suite) version 3.3.0. Villus measurements (mm) were taken with intact lamina and were based on length from the villus apex to the villus crypt junction and width across the base of the villus crypt [25]. For morphometry analysis, 40 measurements from different fields of 12 samples per treatment were considered. 

For microbiota analysis, 2 birds from each replicate (10 per treatment) were selected and taken from each replicate. Cecal microbial DNA was isolated using the Hexadecyl trimethyl ammonium bromide (CTAB)/Sodium dodecyl sulfate (SDS) method and quantified using the Qubit@ 2.0 Fluorometer (Paiseno Biological Technology Co., LTD, Shanghai, China). The V4 region of the 16S ribosomal RNA (16S rRNA) gene was then amplified with 515 F and 806 R primers with the sequences of 5′-GTGCCAGCMGCCGCGGTAA-3′ and 5′-GGACTACHVGGGTWTCTAAT-3′. DNA samples were quantified and then the V4 hypervariable region of the 16S rDNA was amplified. The final amplicon pool was evaluated using the GeneJETTM Gel Extraction Kit (Paiseno Biological Technology Co., LTD, Shanghai, China). Single-end reads were generated with the Ion S5 TM XL platform and filtered using the default parameters and sequenced via Illumina Miseq/Novaseq platform. To investigate the diversity of the cecum microbiota, alpha diversity analysis was made by using the OUT table. Diversity indexes (Chao 1 index, Simpson index, Pielou E index, Shannon index) were calculated. Sequence data analyses were mainly performed using QIIME and R packages (v3.2.0). OTU-level ranked abundance curves were generated to compare the richness and evenness of OTUs among samples. Beta diversity analysis was performed to investigate the structural variation of microbial communities across samples using UniFrac distance metrics.

### 2.7. Statistical Analysis 

The experimental design was a 2 × 2 factorial design, and the main factors were the source of Se and the level of EM. Data were analyzed by two-way ANOVA using SPSS 22.0 (SPSS. Inc., Chicago, IL, USA), which included the main effects of Se source, EM level, and their interaction (Se source × EM level). Tukey’s multiple range test was used to analyze the differences. All data were further subjected to one-way ANOVA. When overall differences were significant, the differences were tested by Duncan’s multiple-range test (SPSS 22.0). The significance of the difference was determined using *p* < 0.05. The results are presented as the mean values and SEM. We performed PCoA analysis at the phylum and genus levels using the vegan package in R 4.0.3 software. The relative abundances of microbial communities and alpha diversity variables (Chao 1 index, Simpson index, Pielou E index, Shannon index) were analyzed via the Student’s *t*-test using R 4.0.3 software.

## 3. Results

### 3.1. Growth Performance

The effects of Se and probiotics on broiler growth performance from 0 to 70 days are shown in Table 2. The F/G of the I-Se group at Starter (0–35d) was significantly greater (*p* < 0.05) than other groups. Meanwhile, the F/G ratio of probiotic diets was lower (*p* < 0.05) than in non-probiotic groups. There was no significant (*p* > 0.05) difference in ADG or ADFI among groups. On growth performance, there was no interaction (*p* > 0.05) between Se source and probiotics.

### 3.2. Tissue Se Content

As shown in Table 3, Se concentrations in the liver, pancreas, breast muscles, thigh muscles, and serum differed significantly (*p* < 0.05). Meanwhile, when compared to the I-Se group, dietary organic Se increased (*p* < 0.05) Se concentrations in the liver, pancreas, breast muscles, and thigh muscle. Furthermore, diets supplemented probiotic foods had increased (*p* < 0.05) Se concentrations in the pancreas, thigh muscle, and serum than the diet without probiotics. Furthermore, significant interactions (*p* < 0.05) between Se sources and probiotics in diets were observed on Se concentrations in the pancreas, thigh muscle, and serum.

### 3.3. Antioxidant Activities 

The effects of Se and probiotics on antioxidant activity were shown in Table 4. There was a significant difference in the activity of T-AOC and GSH-PX rather than SOD or MDA across the groups. Organic Se enhanced (*p* < 0.05) T-AOC activity as compared to inorganic Se diets. When probiotics were given to broiler diets, the activity of T-AOC and GSH-PX was markedly improved (*p* < 0.05) compared to the diet without probiotics. However, there was no interaction (*p* > 0.05) between Se source and probiotics on broiler antioxidant activity. 

### 3.4. Intestinal Morphometry

As demonstrated in Table 5, dietary Se and probiotics had a significant (*p* < 0.05) effect on the crypt depth and villus height/crypt depth ratio of the duodenum and the villus height of the jejunum in all groups. There was no significant (*p* > 0.05) difference in intestinal morphometry between inorganic Se and organic Se in broilers. Meanwhile, probiotic-supplemented diets improved (*p* < 0.05) the V/C of the duodenum and villus height, as well as the V/C of the jejunum. Interestingly, there was a significant interaction (*p* < 0.05) between Se source and probiotics on the crypt depth of the duodenum and the villus height of the jejunum in birds.

### 3.5. Effects of Se and Probiotics on the Composition of the Cecal Microbiota

#### 3.5.1. Alpha Diversity Analysis

The original off-machine data of high-throughput sequencing is initially screened based on sequence quality, and problem samples are requested and supplemented. The primer fragments of the sequence are then removed, and the sequence of unmatched primers is discarded for quality control, denoising, and steps such as splicing and de-chimerism. In total, there were 1,813,224 high-quality 16S rRNA sequences and 7049 OTUs with similarity greater than 97% reads from the 40 samples (Table 6). 

To determine whether there were any differences in alpha diversity (diversity within the community) among the treatment groups, we performed an alpha diversity analysis. As shown in Table 7, there were numerical differences in the alpha diversity among the groups when measured by the observed species richness (Pielou E index and Chao 1 index) and diversity (Shannon and Simpson indices), but none of them were statistically significant (*p* > 0.05). 

#### 3.5.2. Beta Diversity Analysis 

We also performed beta diversity analysis to determine the differences in community structure among the treatment groups using four different metrics (weighted UniFrac, unweighted UniFrac, Jaccard, and Bray–Curtis) for the measurement of beta diversity. However, none of them showed statistical differences among the four treatment groups at q < 005, which is also reflected in the PCoA plots generated based on the weighted UniFrac distance metric (Figure 1A) and the unweighted UniFrac distance metric (Figure 1B).

#### 3.5.3. Taxonomic Assignments



*Taxonomic Assignment at the Phylum Level*



Dietary Se and probiotics supplementation had no effect on broiler cecal microbial composition at phylum level (*p* > 0.05). There were 19 identified phyla of cecal bacteria, and only those with relative abundance exceeding 0.1% of the total are shown in Figure 2A. The predominant phylum in all treatments was *Bacteroidetes*, *Firmicutes*, *Proteobacteria*, and *Actinobacteria* in descending order of relative abundance (Table 8). 



*Taxonomic Assignment at the Genus Level*



There were 158 identified genera of cecal bacteria, and only those with relative abundance exceeding 0.1% of the total are shown in Figure 2B. The predominant genus in all treatments was *Barnesiellaceae*, *Bacteroides*, *Clostridiales*, *Ruminococcaceae*, *Barnesiella*, *Blautia*, *Lachnospiraceae*, *Lactobacillus*, and *Faecalibacterium* (Table 9). The O-Se group and O-Se + pros group significantly increased (*p* < 0.05) the relative abundance of *Barnesiella* in the cecal content as compared to the I-Se and I-Se + pros groups. Similarly, the dietary Se source had a significant effect on the relative abundance of *Lactobacillus* (*p* < 0.05). Besides, significant interactions between Se sources and probiotics in diets on the relative abundance of *Barnesiella* in cecal content were observed (*p* < 0.05).

## 4. Discussion

According to research on the effect of adding Se on broiler growth performance, adding organic Se was generally greater than adding inorganic Se [26]. This was consistent with our findings, which showed that the F/G of the O-Se group was lower at the Starter (0–35 d). Organic Se in the diet is more beneficial to broiler growth performance. However, this positive effect disappeared at Finisher (36–70 d) in this study, which may be related to the selenium requirement of broilers at Finisher (36–70 d) which was higher than that at the Starter (0–35 d). The supplemental concentration does not meet its requirement, which is also a deficiency of this study. Meanwhile, probiotics have been shown to promote growth in previous studies [27,28,29]. Other research findings indicated that EM addition in chicken feed had no significant influence on mortality, feed conversion ratio (FCR), or weight gain [30]. In the current study, probiotics reduced the F/G of birds at the Starter (0–35 d) investigation. This discrepancy might be attributed to variances in bacteria addition amounts, bacterium type, and the bird’s living habitat. Furthermore, our results indicated that Se and probiotics had no interaction effects on broiler growth performance at any period. 

In our research, dietary supplement with organic Se raised the deposition of Se in the liver, pancreas, breast muscles, and thigh muscle compared to inorganic administration. Gul et al. [31] provided similar results when adding yeast Se and sodium selenite to chickens. When methionine is limited or broken down with the release of Se, yeast selenium, the organic form of Se, can be stored in a protein pool and transported via another pool. The inorganic form (sodium selenite) enters the pool directly, all of the Se is utilized as GSH-Px for selenoprotein synthesis, and the excess Se is eliminated [32,33]. Thus, the bioavailability of Se is determined not only by its absorption in the gastrointestinal tract but also by its conversion into a physiologically active form [34,35]. Meanwhile, we found that birds dietary with probiotic had greater Se concentrations in the liver, pancreas, thigh muscle, and serum than those diet without probiotics in our study. Several studies on the effect of Se-rich probiotics on Se deposition in tissues contributed to the support for our findings [9,27], although their study could not clarify whether this effect was caused by Se or probiotics. Significant interactions between Se sources and probiotics on Se concentrations in the pancreas, thigh muscle, and serum were observed in the current study, indicating that Se and probiotics have positive synergistic effect on broiler tissue Se content. 

The antioxidant system of chickens is regulated by numerous important enzymes, including T-AOC, GSH-Px, SOD, and MDA [36]. The antioxidant effect of selenium is generally achieved by GSH-Px, and selenium is also a component of its active center element. Additionally, Se also was required for the proper functioning of an antioxidant enzyme that scavenges free radicals in the body during normal metabolism [37,38,39]. In the current study, our results revealed that organic Se significantly increased the activity of T-AOC but had no significant effect on GSH-Px, SOD, or MDA, and similar results were reported by others [35]. However, there are also some studies that showed different results—that organic Se did not affect the activity of T-AOC but increased the activities of other antioxidant enzymes [37,39]. Different results may be due to different breeds and different breeding stages. Meanwhile, in the present research, probiotic implementation mediated a significant increase in T-AOC and GSH-Px activities, suggesting a strong antioxidant potential. Several studies have found enhanced antioxidant enzyme (T-AOC and GSH-Px) levels in chickens administered various probiotics [14,22,40,41,42]. In addition, in the present study, we found that T-AOC and GSH-Px activities in the O-Se + pros group were the highest among the four groups, whereas there was no interaction between Se and probiotics on the antioxidant activities of broilers in our study, suggesting that selenium combined with probiotics has more potential to improve the antioxidant status of broilers than selenium alone. The reason for the insignificant interaction effect may be related to the proportion level of the two, the form of selenium, and the type of EM bacteria.

In the present study, the addition of Se along with probiotics had a positive effect on the histomorphological properties of the broilers’ gut. Intestinal villi are covered structurally by the intestinal epithelium, with a continuous layer of myofibroblasts beneath the epithelium that maintains epithelial renewal and defensive mechanisms [43]. Crypts play a role in epithelial cell proliferation by generating defensins and dendritic material [44]. Our results showed there was no significant difference in intestinal morphometry between inorganic Se and organic Se in broilers. However, previous studies have reported that organic Se supplementation form was associated with increased intestinal villus height compared to control and sodium selenite-fed chickens [45]. This could be because the concentration of organic and inorganic selenium is beyond the small intestine’s efficient absorption range, exhibiting the superiority of organic selenium. Our research showed that the inclusion of probiotics enhanced the V/C of the duodenum, villus height, and V/C of the jejunum, as well as the villus height of the ileum, which may improve the villus and crypt morphology, hence boosting gut health. Additionally, there is an interaction between the Se source and probiotics on the crypt depth of the duodenum and the villus height of the jejunum of birds.

Several studies have shown that Se and probiotics used in the chicken industry may modulate intestinal microbiota structure by preventing pathogenic bacterial development and stimulating the growth of beneficial bacteria, hence improving the intestinal micro ecological environment [46,47,48,49,50]. In the current study, we did not observe α-diversity discrepancy conducted on cecal content, and no significant effects were found at the broiler phylum level. Many studies have shown that the microbial diversity of the chicken microbiota is relatively lower compared to the intestinal microbiota of other animals, which is attributed to the rapid transit of food through the digestive system, with short retention times [51]. Microbial diversity increases during chicken development, peaking around day 14 in the foregut and then remaining stable or decreasing slightly thereafter [52]. Data from previous studies suggest that the composition of microbiome is more likely to influenced by age rather than treatment [53]. This reasonably explained the absence of significant differences in α-diversity and phylum levels detected in our results. Interestingly, organic Se suggests improving the proliferation of *Barnesiella* and *Lactobacillus* in the cecal at the genus level. *Lactobacillus* is widely recognized for improving gut function and health [54], and *Barnesiella* is a potent immunomodulator capable of preventing the colonization of pathogenic antibiotic-resistant bacteria and combating pathogen overgrowth in the intestine through non-metabolic pathways [55,56]. *Barnesiella* was a member of the family *Porphyromonadaceae* under the phylum *Bacteroidetes*, and its main end products were butyric and iso-butyric acids, with minor quantities of succinic, propionic, and acetic acids generated as well [57]. This suggest that compared with inorganic selenium, the improvement of intestinal health of broilers by using yeast selenium may be related to the increase of *Lactobacillus* and *Barnesiella* abundance. However, the addition of EM bacteria in this experiment did not significantly affect the composition of cecal microorganisms in broilers, which may indicate that organic selenium plays a stronger role in adjusting microbial composition and function than EM bacteria when selenium and probiotics are combined. Therefore, the effect of EM bacteria is not reflected. However, significant interactions between Se sources and probiotics in diets on the relative abundance of *Barnesiella* were observed in the cecal, which is a clue to demonstrate the benefits of dietary Se combined with probiotics for the gut health of broilers.

## 5. Conclusions

In conclusion, the inclusion of 0.2 mg/kg Se and 0.5% probiotics has a synergistic effect on enhancing broiler growth performance, tissue Se content, antioxidant capacity, encouraging intestinal development, and regulating the composition of intestinal flora, especially the combination of organic selenium and probiotics. Furthermore, these findings suggest that Se sources and probiotics had interaction effects on Se concentrations in the pancreas, thigh muscle, serum, the crypt depth of the duodenum and villus height of the jejunum of birds, and the abundance of *Barnesiella* in the cecal, whereas Se sources and probiotics had no significant interactions on broiler growth performance, antioxidant activities, and intestinal microbial diversity indices. These findings provide an effective method of nutrient combination addition to improve the performance of three yellow chickens.

## Figures and Tables

**Figure 1 animals-12-01086-f001:**
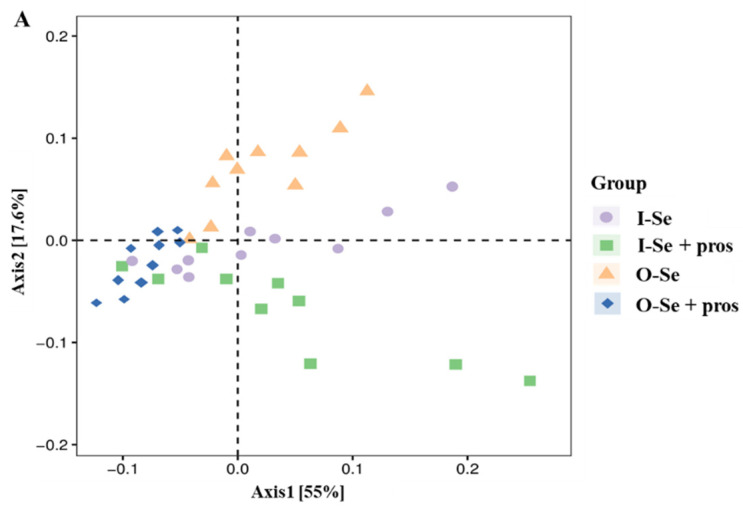
The weighted UniFrac distance metric (**A**) and unweighted UniFrac distance metric (**B**) of the cecal content of broilers fed with I-Se, I-Se + pros, O-Se, O-Se + pros (*n* = 10).

**Figure 2 animals-12-01086-f002:**
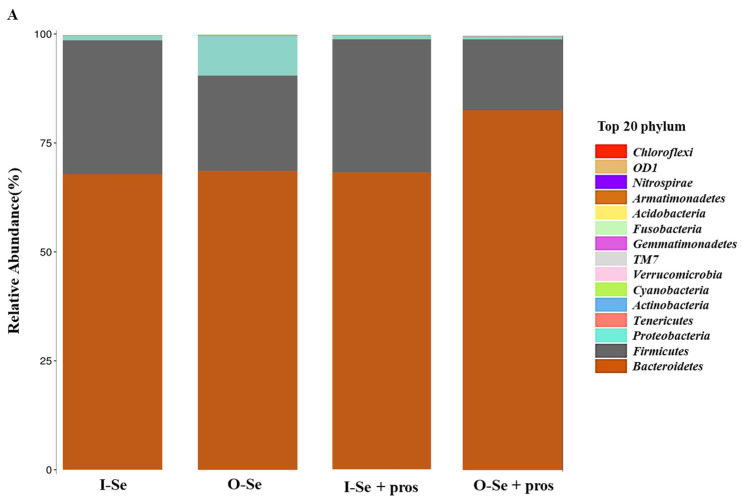
Bacterial composition at the phylum level (**A**) and genus level (**B**) in the cecal content of broilers fed with I-Se, I-Se + pros, O-Se, O-Se + pros (*n* = 10).

**Table 1 animals-12-01086-t001:** Ingredient and nutrient composition of broiler chicken diets (dry matter basis, %).

Ingredient	Content, %		Nutrient Level ^3^	Content	
	1 to 35 d	36 to 70 d		1 to 35 d	36 to 70 d
Corn	59.00	65.00	DE (Mcal/kg)	3.00	3.12
Soybean meal	30.28	20.00	Crude protein (%)	21.26	16.32
Cottonseed meal	2.00	4.50	Crude lipid (%)	4.35	4.56
Fish meal	3.20	-	Crude ash (%)	6.08	5.93
Wheat bran	-	3.15	Ca (%)	0.97	0.88
Soybean oil	1.45	3.00	Total P	0.72	0.65
Ca(H_2_PO_4_)_2_	1.05	1.10	Available P (%)	0.46	0.36
Limestone	1.50	1.70	Lys (%)	1.11	0.86
Choline chloride	0.10	0.10	Met (%)	0.53	0.41
DL-Met	0.18	0.15	Met + Cys (%)	0.86	0.69
NaCl	0.24	0.20	Thr	0.80	0.69
NaHCO_3_	-	0.10			
Mineral premix ^1^	0.98	0.98			
Vitamin premix ^2^	0.02	0.02			
Total	100.00	100.00			

^1^ The mineral premix provided per kilogram of complete feed: Cu, 1.06 mg; Fe, 84.67 mg; Zn, 29.24 mg; Mn, 4.43 mg. ^2^ The vitamin premix provided per kilogram of complete feed: vitamin A, 6500 IU; vitamin D3, 2000 IU; vitamin E, 16 IU; vitamin K3, 2 mg; vitamin B1, 2 mg; vitamin B2, 5 mg; vitamin B6, 1.6 mg; vitamin B12, 0.015 mg; D-biotin, 0.12 mg; D-pantothenic acid, 10 mg; folic acid, 1 mg; nicotinamide, 20 mg. ^3^ The crude protein, crude lipid, and crude ash were measured value, and others were calculated value.

**Table 2 animals-12-01086-t002:** Effects of selenium and probiotics on growth performance of broiler ^1^.

Items ^2^	Se, mg/kg	Pros, %	IW ^5^, g	FBW ^6^, g	Starter (0–35 days)			Finisher (36–70 days)			Whole Term (0–70 d)		
					ADG ^7^, g	ADF ^8^, g	F/G ^9^	ADG, g	ADFI, g	F/G	ADG, g	ADFI, g	F/G
I-Se	0.2	0	35.80	1909.01	19.23	36.78	1.91 ^ab^	34.14	98.82	2.89	26.61	67.50	2.54
O-Se	0.2	0	35.76	1898.09	19.43	36.69	1.89 ^b^	33.26	96.79	2.91	26.30	66.55	2.53
I-Se + pros	0.2	0.5	35.65	1912.58	18.96	37.19	1.96 ^a^	34.46	97.97	2.85	26.65	67.36	2.53
O-Se + pros	0.2	0.5	35.68	1883.30	19.68	37.32	1.90 ^b^	33.07	96.40	2.92	26.36	66.80	2.54
SEM			0.05	11.10	0.12	0.23	0.01	0.71	0.69	0.22	0.16	0.39	0.01
Main effect means ^3^													
Se-S	Inorganic		35.73	1910.80	19.09	36.99	1.94 ^a^	34.30	98.40	2.87	26.63	67.43	2.53
	Organic		35.72	1890.69	19.56	37.00	1.89 ^b^	33.16	96.60	2.92	26.33	66.67	2.54
	SEM		0.15	33.24	0.32	0.69	0.02	0.90	2.01	0.06	0.48	1.18	0.04
Pros		0	35.78	1903.55	19.32	37.25	1.93 ^a^	33.70	97.81	2.90	26.45	67.02	2.54
		0.5	35.67	1897.94	19.33	36.74	1.90 ^b^	33.76	97.19	2.88	26.51	67.08	2.53
		SEM	0.15	33.24	0.32	0.69	0.02	0.90	2.01	0.06	0.48	1.18	0.04
*p*-value													
Treatment			0.735	0.815	0.171	0.759	0.011	0.375	0.620	0.650	0.843	0.832	0.994
Se-S			0.963	0.405	0.055	0.973	0.001	0.095	0.223	0.324	0.386	0.381	0.884
Pros			0.294	0.814	0.972	0.303	0.022	0.920	0.668	0.664	0.873	0.946	0.942
Se-S × Pros ^4^			0.746	0.206	0.256	0.206	0.063	0.697	0.875	0.516	0.701	0.970	0.816

^1^ Data are means with the SEM derived from ANOVA error mean square for *n* = 10. ^2^ I-Se, basal diet containing 0.2 mg/kg sodium selenite; O-Se, basal diet containing 0.2 mg/kg selenium yeast; I-Se + pros, basal diet containing 0.2 mg/kg sodium selenite and 0.5% probiotics; O-Se + pros, basal diet containing 0.2 mg/kg selenium yeast and 0.5% probiotics. ^3^ Se-S, Main effect of selenium source; Pros, Main effect of probiotics; Se-S + Pros, Main effect of selenium source plus probiotics. ^4^ Interaction between Se source and probiotics level. ^5^ IW, initial body weight; ^6^ FBW, weight at 70 d; ^7^ ADG, average daily weight gain; ^8^ ADFI, average daily feed intake; ^9^ F/G, the ratio of feed: gain. ^a,b^ Means within the same row having different superscripts differ significantly (*p* < 0.05).

**Table 3 animals-12-01086-t003:** Effect of selenium and probiotics on the distribution of selenium content in broiler tissues ^1^.

Items ^2^	Se, mg/kg	Pros, %	Liver, mg/kg	Kidney, mg/kg	Pancreas, mg/kg	Breast Muscles, mg/kg	Thigh Muscle, mg/kg	Serum, mg/L
I-Se	0.2	0	0.62 ^b^	0.70	0.21 ^c^	0.16 ^b^	0.16 ^c^	0.12 ^b^
O-Se	0.2	0	0.79 ^a^	0.72	0.39 ^b^	0.38 ^a^	0.33 ^a^	0.12 ^b^
I-Se + pros	0.2	0.5	0.75 ^a^	0.75	0.18 ^c^	0.16 ^b^	0.26 ^b^	0.19 ^a^
O-Se + pros	0.2	0.5	0.80 ^a^	0.61	0.62 ^a^	0.34 ^a^	0.34 ^a^	0.12 ^b^
SEM			0.02	0.02	0.04	0.03	0.02	0.01
Main effect means ^3^								
Se-S	Inorganic		0.68 ^b^	0.72	0.20 ^b^	0.16 ^b^	0.21 ^b^	0.15 ^a^
	Organic		0.79 ^a^	0.68	0.49 ^a^	0.36 ^a^	0.33 ^a^	0.12 ^b^
	SEM		0.05	0.05	0.04	0.02	0.02	0.02
Pros		0	0.70 ^b^	0.71	0.30 ^b^	0.28	0.24 ^b^	0.12 ^b^
		0.5	0.78 ^a^	0.70	0.37 ^a^	0.25	0.30 ^a^	0.16 ^a^
		SEM	0.05	0.06	0.04	0.02	0.02	0.02
*p*-value								
Treatment			0.004	0.236	<0.001	<0.001	<0.001	0.007
Se-S			0.004	0.160	0.001	0.001	0.001	0.040
Pros			0.041	0.457	0.006	0.202	0.002	0.037
Se-S × Pros ^4^			0.094	0.101	0.001	0.367	0.010	0.022

^1^ Data are means with the SEM derived from ANOVA error mean square for *n* = 10. ^2^ I-Se, basal diet containing 0.2 mg/kg sodium selenite; O-Se, basal diet containing 0.2 mg/kg selenium yeast; I-Se + pros, basal diet containing 0.2 mg/kg sodium selenite and 0.5% probiotics; O-Se + pros, basal diet containing 0.2 mg/kg selenium yeast and 0.5% probiotics. ^3^ Se-S, Main effect of selenium source; Pros, Main effect of probiotics; Se-S + Pros, Main effect of selenium source plus probiotics. ^4^ Interaction between Se source and probiotics level. ^a,b,c^ Means within the same row having different superscripts differ significantly (*p* < 0.05).

**Table 4 animals-12-01086-t004:** Effects of selenium and probiotics on serum antioxidant capacity of Broilers ^1^.

Items ^2^	Se, mg/kg	Pros, %	T-AOC ^5^, U/mL	GSH-Px ^6^, U/mL	SOD ^7^, U/mL	MDA ^8^, nmol/mL
I-Se	0.2	0	11.83 ^c^	761.40 ^b^	191.36	3.25
O-Se	0.2	0	14.64 ^bc^	738.30 ^b^	192.00	3.32
I-Se + pros	0.2	0.5	16.44 ^ab^	914.40 ^a^	199.27	2.40
O-Se + pros	0.2	0.5	19.86 ^a^	953.25 ^a^	173.60	3.20
SEM			0.87	28.87	3.80	0.19
Main effect means ^3^						
Se-S	Inorganic		14.13 ^b^	837.90	194.33	2.83
	Organic		16.96 ^a^	833.83	185.10	3.26
	SEM		1.70	57.45	10.26	0.53
Pros		0	13.23 ^b^	749.85 ^b^	191.68	3.29
		0.5	17.96 ^a^	931.67 ^a^	186.44	2.80
		SEM	1.80	60.93	10.26	0.53
*p*-value						
Treatment			0.004	0.005	0.193	0.296
Se-S			0.024	0.853	0.110	0.079
Pros			0.001	0.001	0.483	0.096
Se-S × Pros ^4^			0.808	0.471	0.095	0.056

^1^ Data are means with the SEM derived from ANOVA error mean square for *n* = 10. ^2^ I-Se, basal diet containing 0.2 mg/kg sodium selenite; O-Se, basal diet containing 0.2 mg/kg selenium yeast; I-Se + pros, basal diet containing 0.2 mg/kg sodium selenite and 0.5% probiotics; O-Se + pros, basal diet containing 0.2 mg/kg selenium yeast and 0.5% probiotics. ^3^ Se-S, Main effect of selenium source; Pros, Main effect of probiotics; Se-S + Pros, Main effect of selenium source plus probiotics. ^4^ Interaction between Se source and probiotics level. ^5^ T-AOC, total antioxidative capacity; ^6^ GSH-Px, glutathione peroxidase; ^7^ SOD, superoxide dismutase; ^8^ MDA, malonaldehyde. ^a,b,c^ Means within the same row having different superscripts differ significantly (*p* < 0.05).

**Table 5 animals-12-01086-t005:** Effects of selenium and probiotics on the morphology of duodenum, jejunum, and ileum in Broilers ^1^.

Items ^2^	Se, mg/kg	Pros, %	Duodenum			Jejunum			Ileum		
			V ^5^, µm	C ^6^, µm	V/C ^7^	V, µm	C, µm	V/C	V, µm	C, µm	V/C
I-Se	0.2	0	1208.20	255.87 ^a^	4.80 ^b^	885.37 ^b^	135.23	6.55	878.51	167.50	5.27
O-Se	0.2	0	1188.05	198.12 ^b^	6.05 ^ab^	1155.08 ^a^	150.87	7.85	982.20	167.53	5.87
I-Se + pros	0.2	0.5	1336.74	184.87 ^b^	7.26 ^a^	1306.01 ^a^	144.01	9.09	1001.96	153.33	6.59
O-Se + pros	0.2	0.5	1387.05	222.78 ^ab^	6.37 ^ab^	1206.51 ^a^	143.89	8.40	973.29	165.49	6.04
SEM			38.51	10.92	0.38	58.69	2.64	0.39	19.40	3.37	0.22
Main effect means ^3^											
Se-S	Inorganic		1272.47	220.37	6.03	1095.69	139.62	7.82	940.23	160.42	5.93
	Organic		1287.55	210.45	6.21	1185.94	146.68	8.18	977.74	166.51	5.96
	SEM		61.70	11.38	0.44	102.38	7.39	0.87	43.30	9.47	0.57
Pros		0	1200.14	232.77	5.30 ^b^	993.25 ^b^	141.49	7.07 ^b^	930.36	167.52	5.57
		0.5	1356.87	200.04	6.90 ^a^	1256.26 ^a^	143.95	8.75 ^a^	987.62	159.41	5.95
		SEM	75.57	13.94	0.55	91.58	6.61	0.78	43.30	9.47	0.57
*p*-value											
Treatment			0.232	0.021	0.042	0.013	0.279	0.065	0.082	0.427	0.216
Se-S			0.834	0.465	0.724	0.255	0.162	0.622	0.255	0.390	0.955
Pros			0.055	0.119	0.032	0.011	0.861	0.033	0.098	0.261	0.099
Se-S × Pros ^4^			0.628	0.009	0.075	0.031	0.156	0.132	0.063	0.392	0.189

^1^ Data are means with the SEM derived from ANOVA error mean square for *n* = 10. ^2^ I-Se, basal diet containing 0.2 mg/kg sodium selenite; O-Se, basal diet containing 0.2 mg/kg selenium yeast; I-Se + pros, basal diet containing 0.2 mg/kg sodium selenite and 0.5% probiotics; O-Se + pros, basal diet containing 0.2 mg/kg selenium yeast and 0.5% probiotics. ^3^ Se-S, Main effect of selenium source; Pros, Main effect of probiotics; Se-S + Pros, Main effect of selenium source plus probiotics. ^4^ Interaction between Se source and probiotics level. ^5^ V = Villus height; ^6^ C = Crypt depth; ^7^ V/C = Villus height: Crypt depth. ^a,b^ Means within the same row having different superscripts differ significantly (*p* < 0.05).

**Table 6 animals-12-01086-t006:** Statistics of sample effective sequence and OTU number ^1^.

Items ^2^	Effective Sequence	OTU	Unique OTU
I-Se	163,325	2362	1135
O-Se	149,415	2515	1308
I-Se + pros	121,335	2858	1557
O-Se + pros	170,332	2689	1342

^1^ Data are means with the SEM derived from ANOVA error mean square for *n* = 10. ^2^ I-Se, basal diet containing 0.2 mg/kg sodium selenite; O-Se, basal diet containing 0.2 mg/kg selenium yeast; I-Se + pros, basal diet containing 0.2 mg/kg sodium selenite and 0.5% probiotics; O-Se + pros, basal diet containing 0.2 mg/kg selenium yeast and 0.5% probiotics.

**Table 7 animals-12-01086-t007:** Intestinal microbial α-diversity indices of chickens after dietary selenium and probiotics supplementation ^1^.

Items ^2^	Se, mg/kg	Pros, %	Chao 1 Index	Simpson Index	Shanon Index	Pielou E Index	Observed_Species
I-Se	0.2	0	965.99	0.56	3.70	0.37	865.07
O-Se	0.2	0	1009.06	0.61	4.08	0.40	937.60
I-Se + pros	0.2	0.5	1117.96	0.56	3.77	0.37	990.80
O-Se + pros	0.2	0.5	1124.41	0.48	3.26	0.32	986.37
SEM			75.57	0.05	0.29	0.03	67.66
Main effect means ^3^							
Se-S ^3^	Inorganic		1041.98	0.56	3.73	0.37	927.93
	Organic		1066.73	0.54	3.67	0.36	961.98
	SEM		55.80	0.05	0.37	0.03	50.24
Pros		0	987.53	0.59	3.89	0.38	901.33
		0.5	1121.18	0.52	3.51	0.34	988.58
		SEM	55.80	0.05	0.37	0.03	50.24
*p*-value							
Treatment			0.438	0.584	0.737	0.682	0.590
Se-S			0.762	0.760	0.899	0.879	0.645
Pros			0.129	0.344	0.488	0.414	0.254
Se-S × Pros ^4^			0.822	0.357	0.414	0.402	0.603

^1^ Data are means with the SEM derived from ANOVA error mean square for *n* = 10. ^2^ I-Se, basal diet containing 0.2 mg/kg sodium selenite; O-Se, basal diet containing 0.2 mg/kg selenium yeast; I-Se + pros, basal diet containing 0.2 mg/kg sodium selenite and 0.5% probiotics; O-Se + pros, basal diet containing 0.2 mg/kg selenium yeast and 0.5% probiotics. ^3^ Se-S, Main effect of selenium source; Pros, Main effect of probiotics; Se-S + Pros, Main effect of selenium source plus probiotics. ^4^ Interaction between Se source and probiotics level.

**Table 8 animals-12-01086-t008:** Bacterial taxonomy within the cecal digesta of broilers at the phylum level (four dominant species) ^1^.

Items ^2^	Se, mg/kg	Pros, %	Bacteroidetes	Firmicutes	Proteobacteria	Actinobacteria
I-Se	0.2	0	67.84	30.70	1.00	0.06
O-Se	0.2	0	68.51	21.88	9.06	0.10
I-Se + pros	0.2	0.5	68.20	30.57	0.83	0.06
O-Se + pros	0.2	0.5	82.60	16.25	0.51	0.17
SEM			7.35	7.20	1.17	0.17
Main effect means ^3^						
Se-S ^3^	Inorganic		68.02	30.63	0.92	0.06
	Organic		75.56	19.06	4.79	0.13
	SEM		13.98	13.00	4.90	0.08
Pros		0	68.18	26.29	5.03	0.08
		0.5	75.40	23.41	0.67	0.11
		SEM	13.98	13.00	4.90	0.08
*p*-value						
Treatment			0.673	0.638	0.304	0.505
Se-S			0.468	0.244	0.296	0.226
Pros			0.486	0.762	0.244	0.539
Se-S × Pros ^4^			0.507	0.772	0.261	0.539

^1^ Data are means with the SEM derived from ANOVA error mean square for *n* = 10. ^2^ I-Se, basal diet containing 0.2 mg/kg sodium selenite; O-Se, basal diet containing 0.2 mg/kg selenium yeast; I-Se + pros, basal diet containing 0.2 mg/kg sodium selenite and 0.5% probiotics; O-Se + pros, basal diet containing 0.2 mg/kg selenium yeast and 0.5% probiotics. ^3^ Se-S, Main effect of selenium source; Pros, Main effect of probiotics; Se-S + Pros, Main effect of selenium source plus probiotics. ^4^ Interaction between Se source and probiotics level.

**Table 9 animals-12-01086-t009:** Bacterial taxonomy within the cecal digesta of broilers at the genus level (nine dominant species) ^1^.

Items ^2^	Se, mg/kg	Pros ^2^, %	*Barnesiellaceae*	*Bacteroides*	*Barnesiella*	*Lactobacillus*	*Ruminococcaceae*	*Clostridiales*	*Lachnospiraceae*	*Faecalibacterium*	*Blautia*
I-Se	0.2	0	55.79	7.21	2.99 ^b^	0.18	3.47	4.24	2.23	1.11	2.29
O-Se	0.2	0	54.66	3.75	7.10 ^a^	1.34	3.08	6.53	0.77	2.55	2.21
I-Se + pros	0.2	0.5	57.86	4.06	3.99 ^b^	0.18	4.4	3.6	1.9	1.84	1.56
O-Se + pros	0.2	0.5	71.51	3.47	5.13 ^a^	2.62	3.68	4.7	1.03	0.91	0.18
SEM			4.85	0.75	0.51	0.44	0.58	0.76	0.33	0.37	0.45
Main effect means ^3^											
Se-S^3^	Inorganic		56.83	5.64	3.49 ^b^	0.18 ^b^	3.94	3.92	2.07	1.47	1.92
	Organic		63.09	3.61	6.12 ^a^	1.98 ^a^	3.38	5.61	0.90	0.91	1.19
	SEM		10.31	1.40	0.54	0.76	1.32	1.60	0.66	0.73	0.87
Pros		0	55.22	5.48	5.05	0.76	3.28	5.38	1.50	1.83	2.25
		0.5	64.69	3.76	4.56	1.40	4.04	4.15	1.47	1.37	0.87
		SEM	10.31	1.40	0.54	0.76	1.32	1.60	0.66	0.73	0.87
*p*-value											
Treatment			0.648	0.274	0.004	0.146	0.909	0.623	0.399	0.424	0.354
Se-S			0.561	0.185	0.001	0.044	0.683	0.321	0.114	0.737	0.428
Pros			0.386	0.253	0.393	0.418	0.577	0.463	0.955	0.552	0.152
Se-S × Pros ^4^			0.494	0.335	0.025	0.423	0.906	0.719	0.666	0.144	0.478

^1^ Data are means with the SEM derived from ANOVA error mean square for *n* = 10. ^2^ I-Se, basal diet containing 0.2 mg/kg sodium selenite; O-Se, basal diet containing 0.2 mg/kg selenium yeast; I-Se + pros, basal diet containing 0.2 mg/kg sodium selenite and 0.5% probiotics; O-Se + pros, basal diet containing 0.2 mg/kg selenium yeast and 0.5% probiotics. ^3^ Se-S, Main effect of selenium source; Pros, Main effect of probiotics; Se-S + Pros, Main effect of selenium source plus probiotics. ^4^ Interaction between Se source and probiotics level. ^a,b^ Means within the same row having different superscripts differ significantly (*p* < 0.05).

## Data Availability

The data presented in this study are available on request from the corresponding author.

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
