# Peer review of "Productive Performance, Serum Antioxidant Status, Tissue Selenium Deposition, and Gut Health Analysis of Broiler Chickens Supplemented with Selenium and Probiotics—A Pilot Study"

_animals, 2022, doi:10.3390/ani12091086_

Round 1
Reviewer 1 Report
The manuscript provided an overview of the effects of dietary selenium and probiotics on growth performance and intestinal development in broiler chickens. This manuscript expanded our knowledge of the combined effects of selenium and probiotic bacteria to enhance growth performance and gut health in broilers. The manuscript is well-written. However, the authors need to address the comments below before publishing
L34: Define probiotics and EM
L39: Do you mean breast muscles rather than chest muscles?
L39: Which part of the leg do you mean? Clarify
L40: In which part of the body did you measure (T-AOC)? Clarify
L40-41: In which segment of the gut are these bacteria?
L41: Replace function with composition
L44: Define V/C
L46: Clarify the interactions
L55: What do you mean by three-yellow chickens? Is this a local breed in China? Clarify
L63: Define poultry
L78: Compared to what?
L117: How many birds are in each group and replicate? Unclear
L122: Cite your previous work
L125: Define Se yeast. Is it a commercial product?
L128: Define yeast
L142: What happened to these parameters?
L145: How did you select 40 chicks from each replicate whereas each replicate has 20 chicks only?
L149: What do you mean?
L187: What do you mean by P. R.?
L190: How many birds did you use for microbiota analysis? Figure 1 shows only 3 birds in each group
L217: Add more details
Reviewer 2 Report
Dear authors,
Despite your good efforts and work, I have to reject your manuscript as I can not detect a control group
Reviewer 3 Report
Productive Performance, Serum Antioxidant Status, Tissue Selenium Deposition and Gut health Analysis of Broiler Chickens Supplemented with Selenium and Probiotics---A Pilot Study
The above manuscript re-submitted after my initial rejection. The revised version improved significantly. However, there are still some lacks to the presentation that I address below. I hope the authors can responds to the comments and address their response in the text accordingly. I turned to be positive after these improvements.
Simple summary
Well presented
Abstract
L41: This is discussion “indicating better microbiota function”. In the abstract Discussing the results are not allowed.
Introduction
Thus, the current study is to ….Thus, the current study aimed to…OR Thus, the aim of this study was to …
M&M
L38: ad libitum should be italic
Results
Table 3: p values in the table should be indicated anyways, so use more decimal points if applicable. 0.000 doesn’t show anything.
Discussions
Some of results are not discussed at all. Please provide few lines of discussion where applicable.
Discussion should be improved particularly in the following paragraphs by speculating the reason behind each phenomenon of increased or decreased results. The authors should lay down the mechanisms behind each result.
L423-437
L454-469
Remember that it is not enough to attribute your results by providing other research statement whether in agreement of disagreement with your result. Authors should contribute their own opinion behind important results.
Conclusion
Well-stated
Round 2
Reviewer 3 Report
No additional comments are required except the authors should proofread their revised manuscript for the use of English language (style punctuations, grammatical issues, etc. by a native scientist in the area.
This manuscript is a resubmission of an earlier submission. The following is a list of the peer review reports and author responses from that submission.
Round 1
Reviewer 1 Report
The manuscript provided an overview of the effects of dietary selenium and probiotics on growth and intestinal performance in broiler chickens. This manuscript expanded our knowledge on the combined effects of selenium and probiotic bacteria to enhance growth performance and gut health in broilers. The manuscript is well-written. However, the authors need to address the comments below before publishing
L29, L30, L3, L47: Cite a reference
L36: Why lower Se concentrations for organic Se are detected in the brain, liver, and breast?
L49: Provide a summary of the literature
L55: What is the sex for these broilers? If mixed-sex, how did you distribute among the groups?
L86-87: Provide more details on methodology and instrumentation
L92-99: Insufficient information on the microbiota analysis. Provide more details on DNA extraction and evaluation, sequencing details and data analysis.
In the results section: Add more results on microbial beta diversity, major phyla distribution among groups, and microbial metabolic pathways
Reviewer 2 Report
Dear authors,
This is really a good work but you should have included a control group of birds and the results should be presented and analyzed along with the ones from this group.
Reviewer 3 Report
Effects of dietary selenium and probiotics supplement on growth performance, tissue selenium content, antioxidant capacity, and gut health in broiler chickens
The above manuscript is poorly written and far away from having scientific merit. The authors did not follow the format of the journal and not used the template to input their content into. There are many grammatical and syntax errors in presenting the text in English. The simple summary brings the reader nowhere. The abstract is far beyond the scientific writing while the treatments are not clearly understood, the level of supplementation is not clear, and the conclusion is not exact. The introduction has lots of flaws and contains a lack of evidence for what the authors aimed to hypothesize. The references are not used in the correct place in the Introduction. Some still lack to provide related citations. I can see flaws and inconsistencies all over the text. With this poor presentation of this work, reading and providing detailed comments is impossible. By no means I can accept this work for publication in the present form. Extensive modifications are required when resubmitting this manuscript in the future. I am sorry that I cannot be positive about this work. However, I encourage the authors to prepare their work in a more scientific flow in future submissions.
